# Is Function Similarity Over-Engineered?
# Building a Benchmark

**Rebecca Saul**[1], **Chang Liu**[2], **Noah Fleischmann**[1], **Richard Zak**[1,3],
**Kristopher Micinski**[2], **Edward Raff**[1,3], **James Holt**[4]
[1]Booz Allen Hamilton, [2]Syracuse University,
[3] University of Maryland, Baltimore County, [4]Laboratory for Physical Sciences
Saul_Rebecca@bah.com, cliu57@syr.edu,
Fleischmann_Noah@bah.com, Zak_Richard@bah.com,
kkmicins@syr.edu, Raff_Edward@bah.com, holt@lps.umd.edu

## Abstract

Binary analysis is a core component of many critical security tasks, including reverse engineering, malware analysis, and vulnerability detection. Manual analysis is often time-consuming, but identifying commonly-used or previously-seen functions can reduce the time it takes to understand a new file. However, given the complexity of assembly, and the NP-hard nature of determining function equivalence, this task is extremely difficult. Common approaches often use sophisticated disassembly and decompilation tools, graph analysis, and other expensive preprocessing steps to perform function similarity searches over some corpus. In this work, we identify a number of discrepancies between the current research environment and the underlying application need. To remedy this, we build a new benchmark, REFUSE-BENCH, for binary function similarity detection consisting of high-quality datasets and tests that better reflect real-world use cases. In doing so, we address issues like data duplication and accurate labeling, experiment with real malware, and perform the first serious evaluation of ML binary function similarity models on Windows data. Our benchmark reveals that a new, simple baseline — one which looks at only the raw bytes of a function, and requires no disassembly or other pre-processing — is able to achieve state-of-the-art performance in multiple settings. Our findings challenge conventional assumptions that complex models with highly-engineered features are being used to their full potential, and demonstrate that simpler approaches can provide significant value.

## 1 Introduction

Binary function similarity detection (BFSD) is the problem of determining whether two binary functions are similar in the absence of source code. BFSD plays a central role in many binary analysis tasks, with downstream applications to reverse engineering [11], malware analysis [8; 27; 29], software component analysis [12; 61], and vulnerability detection [14; 62]. Given its extensive use cases, there has been great interest in finding automated solutions to BFSD, particularly from the security community. However, developing such solutions has proven to be quite challenging; even when the source codes of functions are identical, an unrealistic assumption in real-world scenarios, semantically equivalent functions can have drastically different binary representations due to changes in target architecture, build toolchains, compiler flags, and optimization levels.

BFSD has been studied in systems security, programming languages, and most recently, machine learning [39]. Significant research effort has been spent exploring different function representations, including raw bytes [60], assembly [40; 12], intermediate representations [9], control flow graphs [61;

38th Conference on Neural Information Processing Systems (NeurIPS 2024) Track on Datasets and Benchmarks.

32], data flow [14], and dynamic analysis [46]. In machine learning specifically, similar resources have gone into investigating a variety of architectures for BFSD, with proposals involving convolutional neural networks [34], recurrent neural networks [41], graph neural networks [61; 32; 65], seq2seq encoder-decoders [40], and transformers [46; 64], among others. However, comparably little attention has been paid to developing meaningful benchmarks for evaluating the efficacy of new models, making it challenging to judge the utility of individual contributions — an expected issue without good benchmarks [6].

Lack of data and breadth of evaluations has been a constant issue in BFSD research. According to [35], the median ML model in this space is trained on only 3.7k binaries, and almost none are trained on Windows data, despite that operating system being the primary target for reverse engineering and malware analysis work. Earlier surveys of the BFSD literature, while providing important insight, face the same pitfalls. For example, though [39] takes the vital step of evaluating multiple BFSD models on a common dataset, its test dataset has fewer than $1,000$ binaries in total, and consists entirely of Linux files. The largest BFSD benchmark we are aware of is BinKit [26], which contains $243,128$ binaries. However, as these binaries are built from only 51 GNU packages, they lack diversity in terms of developers, coding styles, project sizes, and project types. Additionally, BinKit does not incorporate any Windows binaries.

In this paper, we address this gap in the machine learning BFSD literature by proposing a new benchmark, REFUSE-BENCH[1]. Together, the size and quality of our datasets allow practitioners to draw meaningful conclusions about the appropriateness of existing BFSD models for a variety of downstream applications.

Our contributions include: **(1)** Datasets chosen to mirror real-world situations in computer security, mitigating six major deficiencies commonly found in prior work that prevent generalization. **(2)** A simple baseline model for BFSD, called Reverse Engineering Function Search (REFuSe), which uses only raw-byte features and a basic convolutional neural network. We release REFuSe under the MIT license. **(3)** Evaluations of REFuSe, and other prominent approaches in BFSD, on our selected datasets. Despite its modest design, REFuSe attains state-of-the-art performance on many tasks, suggesting that more work is needed to leverage higher complexity features and architectures to their fullest potential.

Our paper is organized as follows: in §2, we review prior work in machine learning for binary analysis, with an emphasis on binary function similarity detection. In §3, we present the datasets that comprise our benchmark, and in §4, introduce REFuSe, our new baseline. We detail our experiments and their results in §5, and finally conclude in §6.

## 2 Related Work

In developing REFUSE-BENCH, we discovered six major discrepancies between how datasets were constructed for the BFSD problem, a common problem in malware research and the difficulty in evaluation [45; 24; 25; 43]. While such design choices are reasonable from an initial development perspective, they significantly depart from real-world applications, which will be demonstrated in subsection 5.1 where prior methods do not generalize. We list these key insights that we rectify with REFUSE-BENCH's training and testing sets. **C1. Any binary/project application should have all of its functions in only one of the train/test sets, not both**. Many prior works split one binary into both train and test functions [36; 2; 61], which leaks information about the specifics of the source project. Real-world deployments will be on novel binaries, and so this error will over-estimate performance. **C2. You must check for the same function across binaries**. Most works assume that semantically equivalent functions occur only in the same binary/project [39; 2; 5], but we found this is often violated. In many instances we found code from Stackoverflow or other open-source repositories was copy-pasted into many projects, causing information leakage. **C3. Designers need to be specific on labeling details with code.** The choice of data impacts labeling, which is under-specified in many works. Some do not mention these details [16; 31], are ambiguous without the code [28] or do not fully state how steps are performed [61; 54]. These hinder replicability, fair evaluation, and comparison. **C4. Allow standard compiler optimizations**. In particular, function in-lining is an optimization where the compiler merges one smaller function into another (usually) larger function, forming one final function at compiler time. This requires more care to decide what is "semantically

---

[1] `https://github.com/FutureComputing4AI/Reverse-Engineering-Function-Search`

equivalent" and how evaluation is performed, but disabling this ubiquitous optimization [39; 16] hinders real-world generalization. **C5. Use larger datasets with Windows executables.** Windows has the generally highest need for reverse engineering, and thus BFSD tools, but prior works almost exclusively use a smaller amount of linux binaries - less than 5,500 for [16; 36; 28; 39; 31; 2]. **C6. Do not restrict the search space using information not available at deployment time.** Due to the high computational cost of many prior approaches to binary function similarity search (e.g., BSim takes 28 minutes per file to extract vectors), simplifying assumptions have been made but severely hampered real-world applicability. This includes using a "pool size" of at most $X$ functions to search over, where the true nearest function is forced to be in the set of [32; 39; 54] or limiting searchers to a specific compiler setting[54]. However, it has also been long recognized that real-world usage can not rely on either of these assumptions, as malware and user applications have no debug symbols or source code to inform such information[11]. Using a pool size requires knowing, a priori, the true function — which is the explicit purpose of the BFSD task. For example, on the Common Libraries test set, discussed in Section 3.1.2, reducing the pool size to 100 can increase a model's supposed performance by 53%, yet the smallest program in that test set has 217 functions, and the largest single program has 74,736 (7.5x larger than the largest pool size supoprted by [54]). This means the pool size restrictions prevent a method from reliably searching a single program for a related function and, at best, can search on the order of 50 potential binaries exhaustively. To be applicable to deployment, any binary similarity search measure must be able to scale to hundreds of thousands of programs at a minimum [52; 33; 56; 3; 49], and minimum-viable Anti-Virus scale systems are reported at the 20 million binary range [18]. With an average of 200 functions per binary in Assemblage [35], it is clear that the pool size restriction should be avoided. While such a scale is often impractical for researchers to develop their new techniques, our benchmark provides a more realistic scale and evaluation set that can foster research that is more likely to transfer to practice.

## 2.1 ML approaches to BFSD

Though BFSD has been tackled by researchers in numerous areas, in recent years, methods from machine learning have emerged as the dominant technique in this problem space [39]. A 2022 survey paper by Marcelli et. al [39] highlights two ML models of particular significance: Gemini [61], developed by Xu et. al in 2017, and the graph-matching Graph Neural Network (GNN) from Li et. al (2019) [32]. Gemini computes function embeddings from functions' Attributed Control Flow Graphs (ACFGs), and was the first to combine GNNs with the Siamese neural network architecture. Its publication marked a pivotal moment in the field, as it notably improved on earlier work, established machine learning as a viable approach to BFSD, and inspired several lines of future research. In fact, the GNN from Li et. al., which was published two years after Gemini and was the best performing model evaluated in Marcelli et. al.'s survey, leveraged many of the ideas first introduced in Gemini, including using GNNs on ACFGs.

Since the release of [39], research on machine learning for BFSD has shifted to focus heavily on the transformer architecture and large language model (LLM)-inspired designs for assembly-language models [31; 5; 66]. Much of this work draws from the training paradigm introduced in BERT [10], and adopts masked language modeling (MLM), but *not* next sentence prediction (NSP), as a pre-training task. (NSP cannot be used in the context of compiled code, as proximity is uncorrelated with similarity in this domain.) Several novel tasks have been proposed to replace NSP: these tasks include context window prediction [31], def-use prediction [31], execution language modeling [5], strand-symbolic mapping [5], and knowledge integration [66].

In contrast to generalized assembly model approaches, research that exclusively targets the problem of BFSD tends to eschew novel pre-training and explore other changes to the training regime. The authors of BinShot [2] suggest that MLM alone is sufficient for pre-training, and use a new weighted distance vector with a binary cross entropy loss function. FASER [9] skips pre-training all together, and uses an intermediate language, rather than assembly, as input to their model. A final BFSD-focused model is jTrans [54], which combines instruction semantics and control flow information in a transformer-based architecture. jTrans is distinguished by its unique jump target prediction pre-training task, engineered to augment the model's understanding of jump instructions.

## 2.2 Feature Selection in Binary Analysis

Machine learning has been applied to many problems in binary analysis. In addition to the BFSD approaches discussed in §2.1, numerous methods for provenance identification [44; 22], vulnerability search [63; 50], and malware detection [30; 17; 59; 37] have been proposed in recent years. However, nearly all these methods rely on expensive feature engineering techniques, including disassembly, extraction of control-flow graphs, dynamic analysis, and manual construction of feature vectors by experts with significant domain knowledge. As presented in [47] and [19], such a feature set imposes several limitations on its derivative models. Firstly, the performance of these models can be dependent on the accuracy of external binary analysis tools like IDA Pro [21] and Ghidra [1], which are known to produce errors [4]. Secondly, the desired features can be costly to obtain, requiring proprietary licenses, vast computational resources, and/or rare levels of domain knowledge. Together, these difficulties limit the efficacy, generalizability, and accessibility of these techniques.

To avoid the challenges mentioned above, some authors have attempted to train ML models for binary analysis that operate directly on the raw bytes of an executable. Raff et al. [47] proposed MalConv, a convolutional neural net (CNN) that learned to process raw byte sequences for malware classification to try and adapt the "no feature engineering" strategy to malware. Others have continued the non-trivial effort to adapt NLP methods to binary analysis [19], or incorporate raw byte CNNs into more expensive features [34]. Due to this foundation and simplicity, we adapt MalConv to our benchmark to focus on the data labeling, lower the cost to experiment and observe the gap between byte-based and more complex feature extractors.

## 3 Datasets

Table 1: Overview of the five (test) datasets.

| Dataset | OS | No. Binaries | No. Functions |
|---|---|---|---|
| Assemblage | Windows | $135,975$ | $24,545,694$ |
| MOTIF | Windows | $3,095$ | $2,442,164$ |
| Common Libraries | Windows | $40$ | $106,545$ |
| Marcelli Dataset-1 | Linux | $919$ | $668,400$ |
| BinaryCorp | Linux | $9,675$ | $4,791,673$ |

As §2.1 establishes, many techniques have been proposed for applying machine learning to BFSD. However, it is not yet clear which of these techniques best translates to success in downstream security tasks. Our benchmark aggregates the results of experiments on five datasets (Table 1) to answer this question. The Assemblage dataset, discussed in §3.1, is a collection of Windows binaries compiled by scraping GitHub, and is an order-of-magnitude larger than anything previously seen in the BFSD literature. Due to its size, we split it into a train and test set, and use the training portion to train our own baseline BFSD model, REFuSe (see §4). Additionally, we rely on the MOTIF malware family dataset (§3.1.1) to directly measure the utility of our chosen models for malware analysis. To complement MOTIF, we also curate a dataset of common libraries (§3.1.2) to test on, as recognizing functions from libraries such as these is an important part of reverse engineering and software component analysis. Finally, we include two prominent datasets from the BFSD literature, which are both released under the MIT license, in our benchmark. These are Dataset-1 from Marcelli et. al. [39] and BinaryCorp from jTrans [54].

In the rest of this section, we give more detail on the Assemblage, MOTIF, and Common Libraries datasets, as this study marks the first time they are used in the context of BFSD. For more information on Dataset-1 and BinaryCorp, we refer to the papers cited above, which explore them at length.

### 3.1 REFuSe-Bench

We use the Assemblage [35] dataset/system to create our training data and one of our five tests. Assemblage is a distributed system that crawls repositories of source code, compiles the code it finds in a number of specified configurations, and records extensive file and function-level ground truth metadata about each produced binary – avoiding the need for expensive lifting/re-writing [58]. Assemblage is released under the MIT license; we used Assemblage to crawl 165,706 repositories from GitHub containing C or C++ source code, then compiled this code using the Microsoft Visual Studio Compiler and different architectures, compiler versions, optimizations, and build modes. In sum, we produced $807,133$ binaries in 17 unique configurations, with a mean of 47K and a median of 58K binaries per configuration. These binaries included a total of $263,205,895$ functions. For those interested in working with this dataset, we provide the Assemblage "recipe" that reproduces this build.

We took several steps to curate a train and test set of binary functions from the executables in our Assemblage dataset (C5). First, we partitioned the functions into a train (80%) and test (20%) split based on the source codes they were compiled from, meaning that all functions from all binaries compiled from the same source code (but with different compilation flags; C4) were collocated in the same split (C1). Furthermore, we recognized that functions located in the standard libraries used by nearly every C program were likely to be present in most, if not all, binaries in our dataset (C2). (For example, the function *__raise_securityfailure* is present in over 783K binaries.) This demonstrated that a simple train-test partition by originating source code would still leave significant function overlap between the train and test set. Therefore, we applied an additional filter for common functions, requiring that functions that were found in more than half of the binaries in the Assemblage dataset were restricted to appearing on only one side of either the train or the test split. (The threshold for what counts as a common function could be set lower than 0.5, and we invite future work to explore changes to our dataset selection strategy; C1,2,3.) Finally, in both the train and test sets, we filtered out duplicate functions, identified by hashes of the functions' bytes (C2). In the end, we were left with $97,747,033$ functions in the training set and $24,545,694$ functions in the test set.

We considered several labeling methods for our REFUSE-BENCH training and test datasets, and ultimately judged that since these datasets serve different purposes, they would benefit from different labeling procedures (C3). We elaborate on our labeling practice for the training dataset in §4.2 and the test dataset in §5.

### 3.1.1 MOTIF

The Malware Open-source Threat Intelligence Family dataset (MOTIF) [23] is a dataset of $3,095$ (disarmed) malicious PE binaries from 454 malware families, released under the Booz Allen Public License. As reverse engineering and malware analysis are two of the major downstream uses of BFSD, we felt it was important to incorporate real malware into our benchmark. For our experiments with our baseline, REFuSe, we extracted function bytes from the MOTIF binaries using Ghidra [1]. When benchmarking against other models in the literature, we followed the data extraction and preprocessing methods detailed by their creators. In both instances, we labeled functions with the malware family labels of the binaries they were extracted from. We evaluated models based on their ability to group functions from the same malware family together. This gives us a method to evaluate how well a BFSD method operates against real-world malware that may not be well behaved.

Table 2: Projects and versions in the common libraries dataset.

### 3.1.2 Common Libraries

| Project | Versions | Functions | License |
|---------|----------|-----------|---------|
| abseil | 2020-09-23, 20211102.1, 20230125.0, 20230802.0 | 20,568 | Apache-2.0 |
| cjson | 1.7.15, 2020-09-23 | 217 | MIT |
| glfw3 | 2020-09-23, 3.3.7, 3.3.8 | 1,017 | Zlib |
| libxml2 | 2.11.5, 2020-09-23 | 5,143 | MIT |
| openssl | 2020-09-23, 3.0.3, 3.1.0, 3.1.2 | 74,736 | Apache-2.0 |
| sdl1 | 1.2.15, 2020-09-23 | 1,666 | LGPL-2.1-or-later |
| zlib | 1.2.12, 1.2.13, 1.3, 2020-09-23 | 3,198 | Zlib |

While BFSD has most often been studied in binaries originating from the same source code, but compiled differently, practitioners are also interested in understanding similarity as code evolves over a project's development history. To measure this, we present a dataset consisting of multiple versions of seven different prominent Windows projects: abseil, cjson, glfw3, libxml2, openssl, sdl1, and zlib. These projects were collected from vcpkg, Microsoft's open source package manager, and the versions and licenses of each project chosen for our dataset are shown in Table 2. In total, this dataset consists of 40 binaries. Each binary is compiled in release mode to x64 architecture using version 142 of the Microsoft Visual Studio Compiler, with optimization level $0d$.

We used the Assemblage framework [35] to gather function-level metadata about these binaries. For all experiments, we gave functions the same label if they had the same name, and asked models to match functions across project versions. When evaluating on REFuSe, we leveraged Assemblage to get function boundaries. When benchmarking against prior work, we followed the data processing regimes detailed by their authors.

We note that the seven different projects we have chosen contain dramatically different numbers of functions (see Table 2). Because class imbalance is common in real-world binary function similarity

use cases, we opted not to subsample functions from the larger projects, and instead used all the available data. To measure the effects of this imbalance, in §5.1 we broke down our results on a per-project basis.

# 4 REFuSe: A Simple BFSD Baseline

As discussed in §2.1, most prior work in machine learning for BFSD does not directly leverage a function's binary representation. Instead, these approaches extract higher-level information from binaries, such as disassembly or control flow graphs (CFGs), and train models on these features. Naturally, the neural architectures used in these works are geared specifically towards the structure of the features they exploit, e.g. natural language-inspired techniques for disassembly-based methods, or graph neural networks for those relying on CFGs.

To better understand the impact of sophisticated features and heavyweight architecture choices, we sought to develop a simple baseline model for BFSD. We opted out of feature-engineering all together, and decided to train a model straight from raw bytes. Moreover, instead of the traditional GNNs, RNNs, or Transformer-based networks, we used an adapted version of the MalConv convolutional neural network proposed in [47; 48] (and discussed in §2.2). Our model Reverse Engineering Function Search, or REFuSe, modifies MalConv to extract embeddings rather than binary classifications.

The REFuSe architecture is shown in Figure 1. The model takes in a function's raw bytes as input, and, as in [47; 48], begins by mapping each byte to a learned feature vector. This helps the model to distinguish between nearby byte values instead of treating them as inherently similar, which we know is not true in the binary setting. After the byte embedding layer, we pass the function representation to two 1-D convolution layers, as in MalConv, followed by a temporal max pooling layer. Finally, we apply just a single fully-connected layer to produce an embedding.

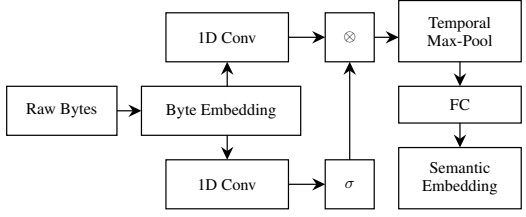

Figure 1: The REFuSe architecture.

## 4.1 The Training Regime: Triplet Learning

We trained REFuSe using triplet learning, a standard metric learning technique for ranked similarity learning with proven success on BFSD tasks [39]. In triplet learning, a Siamese neural network [7] is trained on triplets of examples $(x, x_+, x_-)$, where $x$ is an anchor instance and $x_+$ (positive) is more similar to $x$ than $x_-$ (negative) is to x. That is, for a function $s$ measuring the true similarity of two examples, the set of valid triplets is $T = \{(x, x_+, x_-) \mid s(x, x_+) > s(x, x_-)\}$. The goal of triplet learning is to learn an embedding function $f$ such that for a pre-determined distance metric $d$ and margin of separation $\alpha$, $d(f(x), f(x_+)) + \alpha < d(f(x), f(x_-)) \forall (x, x_+, x_-) \in T$.

In this way, triplet learning incentivizes models to push similar examples together and pull dissimilar examples apart. The idea of using a push-pull mechanism in the loss function was first introduced by [57] in the context of k-nearest neighbor classification. It was later modified to support neural network architectures in [55], and has been utilized in the context of BFSD in [32] and [54]. We used the version of triplet loss popularized in Schroff et. al. [51], with the loss function $L = \frac{1}{|T|} \sum_{(x, x_+, x_-) \in T} \max(d(f(x), f(x_+)) - d(f(x), f(x_-)) + \alpha, 0)$ for an embedding function $f$, distance function $d$, and margin $\alpha$. Schroff et. al. forced their embeddings $f(x) \in \mathbb{R}^n$ to lie on the $n$-dimensional hypersphere, and employed the Euclidean distance as their distance metric. We did not constrain our embeddings, and instead, like [54], used the cosine distance, which is agnostic to vector magnitudes and has greater theoretical support for out-of-distribution detection [42].

In keeping with the recommendations of [51], we preferred to train on semi-hard triplets via online triplet mining. We allowed each function in a mini-batch to serve as the anchor of at most one triplet, and when possible, selected this triplet to be the hardest (biggest contribution to the loss) semi-hard triplet available. If no semi-hard triplets with a given anchor were present in the batch, we selected the easiest (smallest contribution to the loss) hard triplet associated with that anchor. If all triplets associated with an anchor were easy, that anchor did not contribute to our loss.

## 4.2 Building a Training Dataset

The *main* function is an easy example as to why we can't use simple function names as a way of identifying identical functions - many simple names are re-used across programs. For this reason, we take multiple steps to better identify when two functions are considered semantically equivalent. The thresholds specified below were based on manual inspection and random sampling until the results were consistent. Three rules developed below are used to determine function equivalence, and if no rule applied we assumed functions with the same name were equivalent iff they occurred in the same source code/project file.

First, we looked at the sizes of functions (in bytes after compilation) using any given function name. Small functions were associated with basic and near-universal utilities, e.g. those found in a C Runtime library, and were therefore unlikely to be semantically different across binaries. Based on manual inspection and random sampling, we determined if a function was $\leq 25$ bytes in every instance and had the same name, it was labeled as semantically equivalent.

Secondly, we looked at the length of a function's name. We found that long function names tended to describe a function's purpose to such a high level of specificity that even when present in multiple pieces of source code, the semantic effect of that function remained consistent. Thus, function names $\geq 100$ characters were labeled as semantically equivalent.

Third, we reasoned that functions with the same name and roughly the same size were likely to be semantically equivalent. (Small differences in size are expected due to changing build configurations: compiler versions, optimization levels, etc.) We measured the normalized standard deviation (i.e., $\sigma/\mu$) of the function sizes associated with each function name. As a large difference in size is expected between "Debug" and "Release" mode builds, even with all other build configuration variables unaltered, we computed this statistic separately for the two different build modes for each function name. If a function name had a normalized standard deviation of less than 0.05 in both build modes, we did not further partition the labels associated with this function name.

After assigning labels, we took one last step to curate our dataset. Because singleton functions can only provide training signal as the negative member of a training triplet, we downsampled singletons in the training dataset, allowing them to make up at most 5% of all functions. This reduced the size of our training set from $97,747,033$ functions to $82,928,228$ functions. After downsampling, our dataset had a total of $6,801,721$ unique function names. We created subdivided labels based on the originating source code for $759,281$ function names and ended up with a total of $8,897,318$ unique labels in our dataset. We use the standard Mean Reciprocal Rank (MRR) to evaluate the retrieval performance of a model. Due to dataset size, we use approximate nearest neighbor search when possible and return (upper,lower) bound MRR scores. Details on the MRR, bounds, and hyper-parameter settings are in the appendix.

# 5 Experiments and Results

We compared our baseline, REFuSe, against the two representative and SotA models from §2.1: the GNN from [32], and jTrans [54]. These models are not only representative of the major machine learning approaches to binary function similarity detection (graph neural network and transformer-based methods, respectively), but have been also been shown to outperform other work utilizing similar architectures [39; 54]. Both are made available under the MIT license.

As an initial baseline, we use the pre-trained models released by [39] and [54], respectively. To further isolate the effects of model architecture and training data on model performance, we conducted experiments with three additional models. First, we have re-trained the GNN on the Assemblage dataset from scratch (GNN-A), eliminating differences in training data as a factor. The jTrans algorithm does not release its pre-training code, which uses a process different from that of its fine-tuning code; consequently we have only fine-tuned jTrans (jTrans-F) on a sample of the Assemblage data. (The sample consists of 25M functions - compute restrictions prevented us from fine-tuning on the entire Assemblage dataset.) Finally, we experimented with using a basic transformer as a drop-in replacement for the convolutional layers in REFuSe.

In addition to the ML models described above, we also measured REFuSe against BSim, a non-neural tool for function similarity search developed by the Ghidra reverse engineering team [1].[2] BSim works by first disassembling bytes, decompiling the disassembly to an intermediate representation called "p-code", then creating feature vectors from the p-code. It is hand-engineered by experts for finding similar functions and is intended to be invariant to changes in the architectural platform.

Over the course of our research we identified, and endeavored to benchmark against, several other neural models of interest, but were precluded from doing so by a variety of factors. First, as the jTrans paper itself noted[54], the computational cost of most methods is prohibitively exorbitant (e.g., graph matching is NP-hard). Second, most alternatives require proprietary software or do not share the code (and rely on highly technical domain-specific feature engineering). For example, the PalmTree algorithm [31] relies on software called BinaryNinja. Third, the Marcelli work[39] did a large analysis of binary function similarity methods and identified the GNN as the overall most effective. Given the enormous cost of performing this analysis, we believe our current selection of models is well-justified.

**Metrics:** Our primary reported metric is mean reciprocal rank (MRR). A detailed description of the metric is given in Appendix A.2; here, we provide some intuition for understanding the scores. MRR can be interpreted as the average reciprocal of how many $k$ items need to be retrieved to find the first relevant item. For example, an MRR of 0.6 means that, on average, $1/0.6 = 1.67$ items need to be searched to find a relevant match, whereas an MRR of 0.01 would mean searching $1/0.01 = 100$ items on average. Since reverse engineering tasks are only impacted by the first relevant result, MRR is the most directly correlative measure of real-world use, whereas Recall and Precision @k are impacted by the number of relevant items. This is problematic when different items have different denominators, which happens due to the filtering and compiler optimizations that can occur, as discussed in Table 3.

To facilitate comparison with earlier works in the literature, we also report recall@k scores in Appendix C, where recall is calculated as in [54].

**Validating REFuSe-Bench Label Normalization.** As described in §A.1, when training REFuSe, training mini-batches are constructed so that every function has exactly one other function in the batch with its same label (to form an anchor-positive pair). The remaining functions in the batch are then candidates for the negative example in a training triplet. As each function can be part of only one anchor-positive pair, false positives (functions that are semantically different, but mistakenly given the same label) are uniquely damaging to the model, giving extremely erroneous training signal. On the other hand, false negatives (functions that are semantically the same but have different labels) have relatively mild effects. False negative functions are unlikely to be in the same batch (1/8.8M labels), and would almost certainly be a hard-negative (distance $\approx 0$), and so not selected as a semi-hard sample.

While minimizing false positives at the expense of false negatives was advantageous when training REFuSe, the reverse was true when it came to evaluation. In early trials, we noticed that the majority of REFuSe's incorrect predictions were the result of false negatives, where functions with the same name and functionality, but arising from different source codes, were given different labels. Additionally, a significant minority of errors arose when the model matched two functions with the same purpose that operated on different types (e.g. *char* and *wchar_t*).

In light of these observations, we explored the impact of various label normalization schemes on REFuSe's performance. We studied, both separately and together, the effect of two relaxations: allowing functions with the same name to have the same label regardless of originating source code, and allowing functions with the same name to have the same label regardless of parameter types. An example of our normalization scheme is found in Table 3, with the mean MRR achieved on the right.

In Table 5, our primary results table, we report results for REFuSe under the labeling scheme where functions with the same name have the same label, regardless of originating source code, but without parameter type masking (e.g. Table 3, row 3). We chose this labeling scheme because it aligns with the labeling schemes of the other models we evaluated against.

---

[2]see `https://htmlpreview.github.io/?https://github.com/NationalSecurityAgency/ghidra/blob/Ghidra_11.0_build/Ghidra/Configurations/Public_Release/src/global/docs/WhatsNew.html`

Table 3: Normalization schemes for Assemblage function labels.

| Label Norm. | Function 1 | Function 2 | MRR |
|---|---|---|---|
| None | 21991\std::collate<char>::do_compare | 193204\std::collate<wchar_t>::do_compare | 0.088 |
| Mask type | 21991\std::collate<#>::do_compare | 193204\std::collate<#>::do_compare | 0.130 |
| Mask source ID | std::collate<char>::do_compare | std::collate<wchar_t>::do_compare | 0.615 |
| Mask ID & type | std::collate<#>::do_compare | std::collate<#>::do_compare | 0.731 |

Table 4: Mean reciprocal rank (MRR) for each model on each dataset. When Faiss is used for approximate nearest neighbor, we compute (lower, upper) bound MRR scores. **Best results in bold**.

| | Assemblage | MOTIF | Common Libraries | Marcelli Dataset-1 | BinaryCorp |
|---|---|---|---|---|---|
| GNN | (0.263, 0.284) | (0.591, 0.596) | (0.112, 0.138) | (0.050, 0.079) | (0.149, 0.174) |
| GNN-A | (0.267, 0.288) | (0.613, 0.622) | (0.109, 0.135) | (0.054, 0.084) | (0.089, 0.117) |
| jTrans | 0.26 | 0.10 | 0.58 | 0.08 | **0.693** |
| jTrans-F | 0.475 | 0.069 | 0.677 | 0.139 | 0.551 |
| BSim | (0.158, 0.187) | (0.143, 0.150) | (0.383, 0.414) | (0.279, 0.334) | (0.288, 0.322) |
| Naïve Transformer | (0.433, 0.444) | (0.523, 0.530) | (0.496, 0.507) | (0.079, 0.106) | (0.164, 0.187) |
| **REFuSe** | **(0.611, 0.619)** | **(0.678, 0.683)** | **(0.684, 0.691)** | **(0.676, 0.683)** | (0.468, 0.477) |

## 5.1 Results

The complete results of our experiments are displayed in Table 5. Over the Assemblage dataset, REFuSe reached an MRR of 0.61, outperforming both the GNN and jTrans by a score of 0.35.

On MOTIF, the GNN's performance was hindered by being unable to support `.NET` executables and jTrans simply suffered from an inability to generalize to malware. The especially small training set of the GNN further hindered it from the power-law distributed Common Libraries bench, where jTrans did better but still behind REFuSe by $\geq 17\%$. GNN's poor performance on the Common Libraries is further confirmed in appendix Figure 2 to validate that library size was not a confounding factor. jTrans' performance on this task is also marred by the fact that four of the seven libraries in that experiment — abseil, cjson, openssl, and zlib — are also present, in their Linux variants, in jTrans' training data, and so has unfair label leakage in its score.

Remarkably, REFuSe significantly outperforms the GNN on its original test set of Marcelli. (Although higher GNN scores are reported in the original Marcelli paper, the GNN's retrieval power is measured using a curated pool of functions, instead of using the entire test set, as we do here.) Though it appears that jTrans outperforms REFuSe on its test set of BinaryCorp, the results are biased because it is constrained by a "pool-size", the maximum number of functions it can consider. We set the maximum pool size used in the original jTrans paper [54], where their code enforces that the true nearest neighbor is in the pool. Yet, when evaluating REFuSe, we search over all functions in the BinaryCorp test set (4.79M) to find a matching function, not just over $10,000$. This difference in pool size may partially explain the gap between jTrans' and REFuSe's scores, as the jTrans authors themselves demonstrate that the performance of their model decreases as pool size increases [54]. This also limits real-world scalability of jTrans. First, we often never know what optimization levels are used, e.g., the MOTIF dataset is real-world malware with no such information. Second, compiling projects alone creates inequity in unique function counts (see [35], Table 4). Applying filtering and using real optimization like function-inlining further complicates this, making pairwise cross-optimization levels difficult to interpret. For this reason, we do not recommend them as a method of measuring model performance and do not replicate this practice for the evaluations in our paper.

Our first notable result is the strong performance of REFuSe across all experiments. Even though REFuSe does no feature engineering, and uses only a lightweight convolutional neural network, it is the top scoring model on four out of the five datasets we tested. Additionally, its behavior is the most stable of the models we examined, with a range of only 0.22 between its highest and lowest MRRs. Meanwhile, the GNN and jTrans have MRR ranges of 0.53 and 0.59, respectively. While performance on some datasets improves using the GNN-A/jTrans-F variants, which are trained/fine-tuned on Assemblage data, the performance of these models still lags significantly behind that of REFuSe, and helps isolate that these methods are in a sense over-engineered, as they perform worse

even when using the same data. Likewise, BSim's results are much worse than REFuSe's, and are extremely expensive to extract – 24 minutes of CPU compute time per file. Lastly, while the naive transformer model shows promise on the MOTIF and Common Libraries datasets, with an MRR score of 0.53 on both, it is not able to transfer this knowledge to the Linux domain, and we see MRR scores plummet to 0.09 and 0.16 on the Marcelli and BinaryCorp datasets.

From these results, we can infer that (1) byte-based approaches are viable for the BFSD task and (2) more work is needed in feature engineering approaches to scale them to more realistic corpora. In either case, we recommend this suite of test cases as a way to test performance across many scenarios (Assemblage for general binaries, MOTIF for malware, Common Libraries as it is named, Marcelli and BinaryCorp for Linux and historical comparison). We believe that approaches incorporating domain knowledge will eventually outperform REFuSe in the long term, but we argue our results show that such expertise is not being efficaciously used today.

Table 3 demonstrates just how sensitive these results are to labeling schemes. There is a difference of 0.63 in MRR scores between the narrowest and loosest labeling schemes, which, since MRR is measured on a 0 to 1 scale, is extremely significant. To our knowledge, we are the first to investigate the effects of labeling in the context of BFSD, but we hope that our findings encourage others in the binary analysis space to continue this work and to propose thoughtful standards for how function similarity should be defined for a variety of applications. Devising less labor-intensive ways to develop and validate such strategies is a critical open question.

### 5.2 Limitations

In practice, function similarity is not a same/different binary, but a spectrum, where many different factors influence degrees of sameness. For example, functions like *vec<bool>* and *vec<int>* have very similar logic, but very different implementations on the assembly level. Yet many ML training and evaluation methods do not fully capture this nuance. In Table 3, we give an initial analysis of the impact of function labeling based on the labeling criteria we put forth in §5. Many of these criteria are heuristic, and we encourage future work to continue exploring and refining them.

## 6 Conclusion

Binary function similarity detection (BFSD) is a difficult problem that touches many security applications, including reverse engineering, malware analysis, and vulnerability detection. Many ML approaches have been devised to tackle this challenge, but current benchmarks do not mirror the real-world scenarios under which these models would be deployed. We identified five major discrepancies between prior corpora and real-world use, and developed REFuSe-Bench to mitigate these issues. Using a simple CNN over raw bytes, we show superior results compared to current leading BFSD approaches. Our results highlight the need to consider computational scale when using expensive domain feature extractors and how performance can vary across different types of binary data that was not testable in the prior literature. REFuSe-Bench provides the means to accelerate this complex and critical research area.

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

# A    Appendix

## A.1    Experiment Configurations

To obtain our REFuSe model, we initialized the neural net with an embedding size of 8, a window size of 8, a stride size of 8, and an output size of 128. We trained the model for 30 epochs over the Assemblage training dataset, with the model seeing 10M functions per epoch. Functions were divided into batches of 600; for each batch, 300 unique labels were randomly chosen from the training dataset, and then two functions were randomly selected with each label. We used a learning rate of 0.005, and the Adam optimizer with gradients clipped to [-1, 1]. Per the literature [20; 15], we used $\alpha = 0.2$ as the margin for our triplet loss. REFuSe was trained on three Tesla M40s on an internal cluster, and took 4.5 days to train.

The Naïve Attention-based Transformer is a neural net composed of two 1-Dimensional multi-headed dot-product attention encoder layers. These layers are a smaller version of those proposed in [53], initialized with 4 attention heads and a query-key-value dimension of 256. The hidden layer of the internal multilayer perceptron (MLP) has an output dimensionality of 512. The dropout rate for both the MLP and the attention heads is set to 0.1. The model is trained for 32 epochs on three Tesla M40s using the same clipped Adam optimizer, learning rate, and triplet loss as the REFuSe, with the same loss margin, $\alpha$. The model saw 1M functions per epoch in batches of 60, and took 6.5 days to train.

To evaluate the GNN, we used the model checkpoint published by [39] as part of their survey.[3] To evaluate jTrans, we similarly used the fine-tuned model made available on the authors' Github page.[4]

## A.2    Evaluation Procedures

We chose to use mean reciprocal rank (MRR) to measure how models performed on our benchmark. MRR, a popular metric in information retrieval, is used to assess systems which take in queries and return a list of possible responses ordered by likelihood of correctness. Letting $q$ be a query, $L$ be the 1-indexed list returned by $q$, and $c$ be a correctness function, where $c(L[i]) = 1$ if $L[i]$ is a correct response to $q$ and 0 otherwise, $q$ is said to have rank $r$ if the first correct answer in $L$ appears at position $r$. That is, $q$ has rank $r$ if and only if $1 \leq r \leq \text{len}(L)$, $c(L[r]) = 1$, and $c(L[i]) = 0$ for all $1 \leq i < r$. The reciprocal rank of $q$ is defined to be $\frac{1}{r}$, and the mean reciprocal rank is the average of the reciprocal ranks for every query $q \in Q$. The upper bound on MRR is 1.0 (a correct answer is always in the first position in $L$), whereas the lower bound on MRR is 0.

In the context of BFSD, $q$ is a query function and $L$ is a list of neighboring functions (embeddings), ordered from nearest to farthest. Due to the large size of our datasets, we used the Hierarchical Navigable Small Worlds [38] approximate nearest neighbor index from Faiss [13] to compute the 30 nearest neighbors to each query function. When no match was found within the first 30 neighbors, we assigned that query an upper bound reciprocal rank of $\frac{1}{31}$ and a lower bound reciprocal rank of 0.

In Section 5.1, we reported the lower and upper bound MRR for the experiments that used our evaluation code. For benchmarking experiments that utilized open-source code from other authors, we reported a single MRR value, keeping with their practice. In particular, when conducting experiments with jTrans, we chose to use the evaluation code published by its authors, as integrating our own code into their codebase was not straightforward. jTrans supports evaluation over multiple pool sizes; in Section 5.1, we report results for pool size $10,000$, as a larger pool size more closely mimics the evaluation methods of the other models. (In our evaluation, the pool is the entire dataset, but we are not limited to having only one function matching the query function in each pool.)

# B    GNN Common Libraries Details

In our results we stated the significant drop in the GNN's performance on the Common Libraires corpus is due to its inability to handle the variety of functions and function sizes in each application. This is important to verify as the actual cause, as the asperity in the project sizes could easily dominate the results and make it unclear which method actually performs best.

---

[3]This model is available at the following link: https://github.com/Cisco-Talos/binary_function_similarity/tree/main/Models/GGSNN-GMN/NeuralNetwork/model_checkpoint_GGSNN_pair.

[4]This model can be downloaded from https://github.com/vul337/jTrans/.

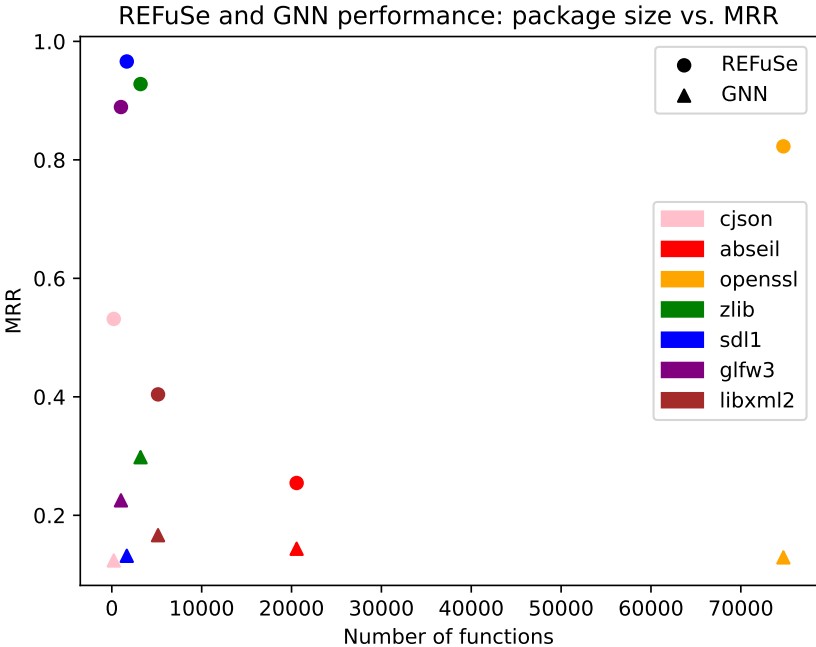

Figure 2: REFuSe and GNN per-package performance. The circles correspond to REFuSe results, while the triangles correspond to the GNN.

We perform this validation in Figure 2, where it can be seen that REFuSe dominates the GNN in performance for each library. Though there are too few libraries to make a definitive conclusion, REFuSe seems to be unfazed by the number of functions in terms of final MRR performance. Yet, the GNN has low performance in all cases and decreases with the number of functions.

# C   Recall@k Results

Table 5: Recall@k for k = 1, 2, 5, 10.

| Model | Dataset | MRR | R@1 | R@2 | R@5 | R@10 |
|---|---|---|---|---|---|---|
| GNN | Assemblage | (0.263, 0.284) | 0.238 | 0.265 | 0.295 | 0.318 |
| GNN | MOTIF | (0.591, 0.596) | 0.499 | 0.592 | 0.709 | 0.782 |
| GNN | Common Libraries | (0.112, 0.138) | 0.085 | 0.104 | 0.135 | 0.159 |
| GNN | Marcelli Dataset-1 | 0.53 | 0.44 | 0.53 | 0.625 | 0.70 |
| GNN | BinaryCorp | (0.149, 0.174) | 0.124 | 0.141 | 0.165 | 0.183 |
| GNN-A | Assemblage | (0.267, 0.288) | 0.236 | 0.264 | 0.296 | 0.320 |
| GNN-A | MOTIF | (0.613, 0.622) | 0.568 | 0.619 | 0.671 | 0.695 |
| GNN-A | Common Libraries | (0.109, 0.135) | 0.085 | 0.106 | 0.137 | 0.161 |
| GNN-A | Marcelli Dataset-1 | 0.14 | 0.065 | 0.115 | 0.2 | 0.33 |
| GNN-A | BinaryCorp | (0.089, 0.117) | 0.079 | 0.091 | 0.109 | 0.123 |
| jTrans | Assemblage | 0.26 | 0.221 | 0.288 | 0.338 | 0.376 |
| jTrans | MOTIF | 0.10 | 0.043 | 0.063 | 0.096 | 0.128 |
| jTrans | Common Libraries | 0.58 | 0.510 | 0.600 | 0.689 | 0.748 |
| jTrans | Marcelli Dataset-1 | 0.08 | 0.068 | 0.076 | 0.105 | 0.132 |
| jTrans | BinaryCorp | 0.693 | 0.625 | 0.700 | 0.775 | 0.826 |
| jTrans-F | Assemblage | 0.475 | 0.347 | 0.427 | 0.531 | 0.613 |
| jTrans-F | MOTIF | 0.069 | 0.030 | 0.040 | 0.050 | 0.083 |
| jTrans-F | Common Libraries | 0.677 | 0.577 | 0.682 | 0.797 | 0.852 |
| jTrans-F | Marcelli Dataset-1 | 0.139 | 0.065 | 0.089 | 0.130 | 0.166 |
| jTrans-F | BinaryCorp | 0.557 | 0.551 | 0.618 | 0.684 | 0.732 |
| BSim | Assemblage | (0.158, 0.187) | 0.143 | 0.164 | 0.175 | 0.181 |
| BSim | MOTIF | (0.143, 0.150) | 0.087 | 0.103 | 0.135 | 0.179 |
| BSim | Common Libraries | (0.383, 0.414) | 0.389 | 0.442 | 0.487 | 0.515 |
| BSim | Marcelli Dataset-1 | (0.279, 0.334) | 0.231 | 0.280 | 0.336 | 0.371 |
| BSim | BinaryCorp | (0.288, 0.322) | 0.251 | 0.291 | 0.329 | 0.354 |
| Naïve Transformer | Assemblage | (0.433, 0.444) | 0.364 | 0.431 | 0.514 | 0.571 |
| Naïve Transformer | MOTIF | (0.523, 0.530) | 0.445 | 0.519 | 0.615 | 0.683 |
| Naïve Transformer | Common Libraries | (0.496, 0.507) | 0.443 | 0.497 | 0.557 | 0.599 |
| Naïve Transformer | Marcelli Dataset-1 | (0.079, 0.106) | 0.054 | 0.077 | 0.108 | 0.130 |
| Naïve Transformer | BinaryCorp | (0.164, 0.187) | 0.127 | 0.160 | 0.206 | 0.241 |
| REFuSe | Assemblage | (0.611, 0.619) | 0.567 | 0.610 | 0.664 | 0.700 |
| REFuSe | MOTIF | (0.678, 0.683) | 0.623 | 0.675 | 0.744 | 0.792 |
| REFuSe | Common Libraries | (0.684, 0.691) | 0.648 | 0.683 | 0.727 | 0.757 |
| REFuSe | Marcelli Dataset-1 | (0.676, 0.683) | 0.629 | 0.679 | 0.727 | 0.752 |
| REFuSe | BinaryCorp | (0.468, 0.477) | 0.380 | 0.426 | 0.655 | 0.674 |

# D jTrans Poolsize Experiments

In the tables below, we show jTrans results for various pool sizes. As can be seen, jTrans continually drops in performance as the pool size increases. Real-world BFSD applications require unbounded pool size, as the purpose of doing binary function similarity search is because one does not know which function(s) are relevant or not, and a pool size requires knowing that the relevant function(s) will be in the pool. This requires information that can not be available at deployment time, and so prevents practical use.

Table 6: Pool size 100.

| Dataset | MRR | Recall@1 | Recall@2 | Recall@5 | Recall@10 |
|---|---|---|---|---|---|
| MOTIF | 0.12162 | 0.06045 | 0.09068 | 0.14861 | 0.20654 |
| Common Libraries | 0.8287 | 0.7623 | 0.82497 | 0.90993 | 0.97771 |
| Marcelli Dataset-1 | 0.25098 | 0.17907 | 0.22953 | 0.30967 | 0.37274 |
| BinaryCorp | 0.95266 | 0.92284 | 0.96708 | 0.990045 | 0.99483 |
| Assemblage | 0.54041 | 0.49128 | 0.52894 | 0.57686 | 0.62166 |

Table 7: Pool size 500.

| Dataset | MRR | Recall@1 | Recall@2 | Recall@5 | Recall@10 |
|---|---|---|---|---|---|
| MOTIF | 0.07952 | 0.0403 | 0.07052 | 0.10579 | 0.14357 |
| Common Libraries | 0.71712 | 0.63231 | 0.71263 | 0.78923 | 0.83983 |
| Marcelli Dataset-1 | 0.22766 | 0.14024 | 0.17759 | 0.23893 | 0.29037 |
| BinaryCorp | 0.88613 | 0.83449 | 0.89732 | 0.95275 | 0.97881 |
| Assemblage | 0.47748 | 0.43778 | 0.47601 | 0.51476 | 0.54302 |

Table 8: Pool size 1000.

| Dataset | MRR | Recall@1 | Recall@2 | Recall@5 | Recall@10 |
|---|---|---|---|---|---|
| MOTIF | 0.08685 | 0.05037 | 0.70529 | 0.10831 | 0.14609 |
| Common Libraries | 0.7197 | 0.58857 | 0.67223 | 0.75673 | 0.80547 |
| Marcelli Dataset-1 | 0.22226 | 0.12383 | 0.15829 | 0.2132 | 0.26053 |
| BinaryCorp | 0.84782 | 0.78946 | 0.85602 | 0.92058 | 0.95579 |
| Assemblage | 0.44986 | 0.40982 | 0.45069 | 0.48973 | 0.5165 |

Table 9: Pool size 5000.

| Dataset | MRR | Recall@1 | Recall@2 | Recall@5 | Recall@10 |
|---|---|---|---|---|---|
| MOTIF | 0.06026 | 0.03275 | 0.04282 | 0.06801 | 0.08816 |
| Common Libraries | 0.58245 | 0.49768 | 0.58774 | 0.68152 | 0.7363 |
| Marcelli Dataset-1 | 0.09861 | 0.0459 | 0.62767 | 0.88162 | 0.1129 |
| BinaryCorp | 0.74619 | 0.67976 | 0.74899 | 0.82216 | 0.87229 |
| Assemblage | 0.37055 | 0.32358 | 0.37534 | 0.42001 | 0.44904 |

