# OpenReview forum: "Is Function Similarity Over-Engineered? Building a Benchmark"
_NeurIPS.cc/2024/Datasets_and_Benchmarks_Track — NeurIPS 2024 Track Datasets and Benchmarks Poster_

### Official Review · Reviewer_ym5A · 2024-07-24
**Interesting contribution with some issues**

**Rating:** 5
**Confidence:** 4

**Review:**

Pros:
1. The paper proposes a simpler model, REFuSe, which achieves state-of-the-art performance on many BFSD tasks, indicating potential improvements by focusing on simpler models.
2. It introduces REFUSE-BENCH, a more realistic and challenging benchmark. This could lead to the development of more effective BFSD models in the future.

Cons:
1. The paper considers 2 baselines: GNN and jTrans. The original papers for GNN and jTrans report higher performance on their datasets compared to the results here (as low as 0.10 for jTrans). The lack of discussion of this discrepancy leaves it unclear if the differences are due to dataset variations or other factors, weakening the claim of REFuSe's superiority.
2. jTrans performs better with smaller pool sizes. The paper doesn't compare REFuSe with jTrans using a smaller pool size, which could be a fairer comparison on different datasets.
3. Given these results, it's difficult to definitively say that REFuSe outperforms all existing models in all tasks. The comparisons with GNN and jTrans seem limited to specific cases and may not generalize to other datasets or scenarios. Including comparisons on more baselines and exploring different scenarios (e.g., including various jTrans pool sizes) would strengthen the paper's contribution.
4. The paper relies solely on MRR for performance evaluation. Including additional metrics, such as AUC or Recall@1, would strengthen the paper's claims.
5. The paper argues for simpler models but could be strengthened by a deeper analysis of why complex models like GNN and jTrans underperform on certain datasets compared to the simpler REFuSe.

**Strengths:**

- The paper contributes to the field by proposing a new model and benchmark, even if further validation is needed.
- The paper introduces a new and apparently simpler model, REFuSe, that achieves state-of-the-t-art performance on many BFSD tasks. This demonstrates that complex features and architectures might not always be necessary for good results.
- The introduction of the new benchmark offers a potentially valuable tool for the field, aiming to be more realistic and challenging to advance the development of effective BFSD models.

**Additional Feedback:**

Figure 1 overlaps with the text and needs improvement. Additionally, table titles should be descriptive enough to explain the contents of the table.

**Clarity:**

The paper appears to be well-structured, with clear introductions to the problem, methodology, results, and discussion sections.

**Correctness:**

Details of the dataset are provided in Section 3, and the evaluation methods along with experiment design are covered in Section 5. All other details are given in the supplementary materials.

**Documentation:**

All the details are provided in the supplementary materials.

**Limitations:**

The authors addressed the potential limitations of their work.

**Opportunities For Improvement:**

See above.

**Relation To Prior Work:**

Yes, the paper introduces a simpler model (REFuSe) using raw bytes and a new benchmark (REFUSE-BENCH) that addresses shortcomings of prior works in both data labeling and benchmark design.

**Summary And Contributions:**

The paper argues that many existing binary function similarity detection (BFSD) models are over-engineered, and that simpler models can achieve state-of-the-art performance. It introduces a new baseline model, REFuSe, which uses only raw bytes as features and a basic convolutional neural network, achieving state-of-the-art performance on BFSD tasks. Additionally, the paper proposes a new benchmark, REFUSE-BENCH, to aid future research in BFSD.

---

> ### Author Rebuttal · Authors · 2024-08-15
>
> **C1**: We discussed factors about jTran's decrease in performance, specifically in lines 359-365, as jTrans' pool of candidate functions is smaller than the set of all functions. The original jTrans code forces the true nearest function to be in the pool, providing the approach an unrealistic advantage in search (i.e., if we already knew the true nearest function to force it into the pool, we had no need to do the search at all). As their results show, performance degrades as the pool size increases. In our case, we have an unrestricted set of all existing functions, which better matches real-world needs, and jTrans' performance drops significantly because of it.
>
> jTrans and GNN also do not perform the normalization steps for cross-project functions and have smaller evaluation sets, making it easier to achieve high performance.
>
> A second issue that we did not discuss in the paper, is the jTrans does not compare all optimization levels at the same time. They compare only two optimization levels at a time, and average the results. This, again, makes the search problem easier by reducing the search space.
>
> To further isolate that GNN and jTrans are, to some degree, over-engineered with respect to their performance, we have conducted two additional experiments. We have created GNN-A, where we re-train the GNN model on the Assemblage data. This decreases performance for most of our test sets, showing that the GNN approach would not be rectified by using the same training data. Please see the all-author rebuttal for details.
>
>
> jTrans did not make their training code available, but they did make a fine-tuning code available (which, upon inspection, is not sufficient to train from scratch and follow all the paper's specified details). Using this we create jTrans-F, which is fine-tuned to the Assemblage data. This improves jTrans' performance on all but the BinaryCorp dataset (which makes sense, BinaryCorp is a case of information leakage that was trained on the same projects), but is still far behind REFuSe's performance on the datasets.
>
>
> **C2**: We will clarify that this is not an experiment that aligns with practitioner use. The limited pool size is an unfair advantage for jTrans relative to real-world use (see above; they put the true nearest neighbor into the pool), where an unbounded number of potential functions to search exists. In addition, as noted above, jTrans limits to searching only pairs of optimization levels --- but in real life we *do not know* the optimization level of any given piece of code. It is these issues that precisely limit the applicability of prior work to real-world deployment and will be clarified in revision. Our tests, using all available test-set data, better match what real-world functionality is needed for a method to be deployable.
>
> Further, because jTrans has a 10k pool *in our experiments*, jTrans has an enormous advantage over REFuSe because it is searching 24.5M test samples, jTrans is comparing to 2450$\times$ fewer samples per search.
>
> **C3**:  Please see our all-author rebuttal, we have added GNN and jTrans versions trained/fine-tuned to Assemblage to eliminate this discrepancy and further show that REFuSe is working more effectively. Please see (C2) response re pooling, jTrans' pool size is an unrealistic limitation that gives jTrans an impossible in real-life advantage over other techniques.
>
> **C4**: Please see the all-author rebuttal for Recall@k metrics which are almost complete. It is clear though that the results are inline with our MRR results. REFuSe performs the best by a significant margin on all but the BinaryCorp dataset, where jTrans performed better but also was explicitly trained for that dataset.
>
> **C5**: Per all-author rebuttal, our new GNN-A and jTrans-F runs strengthen the conclusion that simpler models perform better on these datasets, as the performance of the added models is still far behind that of REFuSe. Furthermore, these experiments show the great asperity between benchmarks even when trained on or fine-tuned to the same data we have used. REFuSe has much higher scores and more stable performance between datasets, which is encouraging for its real-world applicability.
>
> In addition, the BSim results further show that a major reverse engineering tool designed for this task that relies on highly sophisticated expertise is outperformed by REFuSe on these datasets.
>
> > Figure 1 overlaps with the text and needs improvement. Additionally, table titles should be descriptive enough to explain the contents of the table.
>
> Thank you for the typographical catches, we will remediate them!

---

> > ### Comment · Reviewer_ym5A · 2024-08-22
> >
> > Thank you for addressing all the Cons.
> > The authors have presented new results on various datasets and metrics in their rebuttal to support their claims. However, the focus remains on scenarios with larger pool sizes, where jTrans tends to perform better with smaller pool sizes. If the limited pool size is an unfair advantage for jTrans relative to real-world use, it would be valuable to highlight this in the paper, accompanied by relevant use cases. Further exploration and empirical evidence are needed to quantify this effect and demonstrate its relevance to real-world scenarios. A detailed explanation, along with experimental results, would help clarify the claim that “the limited pool size is an unfair advantage for jTrans relative to real-world use”.

---

> > > ### Author Rebuttal · Authors · 2024-08-23
> > >
> > > We are glad that we have answered all of your questions.
> > >
> > > >If the limited pool size is an unfair advantage for jTrans relative to real-world use, it would be valuable to highlight this in the paper, accompanied by relevant use cases.
> > >
> > > We propose adding the below paragraph to the manuscript to help clarify the gap between pool size and real-world needs. We will also add **C6. Restricting the search space using information not available at deployment time** as an additional discrepancy between research and practice.
> > >
> > > Due to the high computational cost of many prior approaches to binary function similarity search  (e.g., Bsim takes 28 minutes per file to extract vectors), simplifying assumptions have been made but severely hampered real-world applicability. This includes using a "pool size" of at most $K$ functions to search over, where the true nearest function is forced to be in the set of $K$ [28,33,41] or limiting searchers to a specific compiler setting [41]. However, it has also been long recognized that real-world usage can not rely on either of these assumptions, as malware and user applications have no debug symbols or source code to inform such information [10]. Using a pool size requires knowing, apriori, the true function --- which is the explicit purpose of the BFSD task. For example, on the Common Libraries test set, reducing the pool size to 100 increases jTran's purposed performance by 53% to an MRR of 0.89. yet the _smallest_ program in that test has 217 functions, and the largest single program has 74,736 (larger than what we can get the jTrans code to work with at all). This means the pool size restrictions prevent a method from reliably searching a single program for a related function and, at best, can search on the order of 50 potential binaries exhaustively.  To be applicable to deployment, any binary similarity search measure must be able to scale to hundreds of thousands of _programs_ at a minimum [A,B,C,D,E], and minimum-viable Anti-Virus scale systems are reported at the 20 million binary range [F]. With an average of 200 functions per binary in Assemblage [30], it is clear that the pool size restriction should be avoided. While such a scale is often impractical for researchers to develop their new techniques, our benchmark provides a more realistic scale and evaluation set that can foster research that is more likely to transfer to practice.
> > >
> > >
> > > Thank you for the recommendation; we believe this will help avoid confusion for other readers.
> > >
> > > Added citations:
> > >
> > > **A.** J. Upchurch and X. Zhou, “Malware provenance: code reuse detection in malicious software at scale,” in 2016 11th International Conference on Malicious and Unwanted Software (MALWARE).  2016
> > >
> > > **B.** Y. Li, S. C. Sundaramurthy, A. G. Bardas, X. Ou, D. Caragea, X. Hu, and J. Jang, “Experimental Study of Fuzzy Hashing in Malware Clustering Analysis,” in 8th Workshop on Cyber Security Experimentation and Test (CSET 15)
> > >
> > > **C.** S. Wehner, “Analyzing Worms and Network Traffic Using Compression,” Journal of Computer Security, vol. 15, no. 3, pp. 303–320,  2007
> > >
> > > **D.** M. Ali, J. Hagen and J. Oliver, "Scalable Malware Clustering using Multi-Stage Tree Parallelization," 2020 IEEE International Conference on Intelligence and Security Informatics (ISI), Arlington, VA, USA, 2020, pp. 1-6, doi: 10.1109/ISI49825.2020.9280546.
> > >
> > > **E.** Edward Raff and Charles Nicholas. A Survey of Machine Learning Methods and Challenges for Windows Malware Classification. In NeurIPS 2020 Workshop: ML Retrospectives, Surveys & Meta-Analyses (ML-RSA). 2020
> > >
> > > **F.** Harang, R.E., & Rudd, E.M. (2020). SOREL-20M: A Large Scale Benchmark Dataset for Malicious PE Detection. Proceedings of the Conference on Applied Machine Learning in Information Security

---

> > > > ### Author Rebuttal · Authors · 2024-08-27
> > > >
> > > > It took some time to run, but we now have the below jTrans results for various pool sizes that will be included in the appendix. As can be seen, jTrans continually drops in performance as the pool size increases. Real-world use requires unbounded pool size as the purpose of doing binary function similarity search is one does not know which function(s) are relevant or not, and a pool size requires knowing that the relevant function(s) will be in the pool. This requires information that can not be available at deployment time, and so prevents practical use.
> > > >
> > > > Pool size of 100
> > > > | Dataset | MRR | Recall@1 | Recall@2 | Recall@5 | Recall@10 |
> > > > |---|---|---|---|---|---|
> > > > | MOTIF | 0.12162 | 0.06045 | 0.09068 | 0.14861 | 0.20654 |
> > > > | CommonLib | 0.8287 | 0.7623 | 0.82497 | 0.90993 | 0.97771 |
> > > > | Marcelli-1 | 0.25098 | 0.17907 | 0.22953 | 0.30967 | 0.37274 |
> > > > | Bincorp-3M | 0.95097 | 0.91953 | 0.96642 | 0.99046 | 0.99547 |
> > > > | Bincorp-26m | 0.95266 | 0.92284 | 0.96708 | 0.990045 | 0.99483 |
> > > > | Assemblage | 0.54041 | 0.49128 | 0.52894 | 0.57686 | 0.62166 |
> > > >
> > > > Pool size of 500
> > > > | Dataset | MRR | Recall@1 | Recall@2 | Recall@5 | Recall@10 |
> > > > |---|---|---|---|---|---|
> > > > | MOTIF | 0.07952 | 0.0403 | 0.07052 | 0.10579 | 0.14357 |
> > > > | CommonLib | 0.71712 | 0.63231 | 0.71263 | 0.78923 | 0.83983 |
> > > > | Marcelli-1 | 0.22766 | 0.14024 | 0.17759 | 0.23893 | 0.29037 |
> > > > | Bincorp-3M | 0.87914 | 0.82146 | 0.89458 | 0.95297 | 0.97733 |
> > > > | Bincorp-26m | 0.88613 | 0.83449 | 0.89732 | 0.95275 | 0.97881 |
> > > > | Assemblage | 0.47748 | 0.43778 | 0.47601 | 0.51476 | 0.54302 |
> > > >
> > > > Pool size of 1000
> > > > | Dataset | MRR | Recall@1 | Recall@2 | Recall@5 | Recall@10 |
> > > > |---|---|---|---|---|---|
> > > > | MOTIF | 0.08685 | 0.05037 | 0.70529 | 0.10831 | 0.14609 |
> > > > | CommonLib | 0.7197 | 0.58857 | 0.67223 | 0.75673 | 0.80547 |
> > > > | Marcelli-1 | 0.22226 | 0.12383 | 0.15829 | 0.2132 | 0.26053 |
> > > > | Bincorp-3M | 0.83574 | 0.76851 | 0.84765 | 0.9215 | 0.95626 |
> > > > | Bincorp-26m | 0.84782 | 0.78946 | 0.85602 | 0.92058 | 0.95579 |
> > > > | Assemblage | 0.44986 | 0.40982 | 0.45069 | 0.48973 | 0.5165 |
> > > >
> > > > Pool size of 5000
> > > > | Dataset | MRR | Recall@1 | Recall@2 | Recall@5 | Recall@10 |
> > > > |---|---|---|---|---|---|
> > > > | MOTIF | 0.06026 | 0.03275 | 0.04282 | 0.06801 | 0.08816 |
> > > > | CommonLib | 0.58245 | 0.49768 | 0.58774 | 0.68152 | 0.7363 |
> > > > | Marcelli-1 | 0.09861 | 0.0459 | 0.62767 | 0.88162 | 0.1129 |
> > > > | Bincorp-3M | 0.71603 | 0.63627 | 0.72075 | 0.8091 | 0.86608 |
> > > > | Bincorp-26m | 0.74619 | 0.67976 | 0.74899 | 0.82216 | 0.87229 |
> > > > | Assemblage | 0.37055 | 0.32358 | 0.37534 | 0.42001 | 0.44904 |

---

### Official Review · Reviewer_4C4P · 2024-07-28
**Review of Submission 1320**

**Rating:** 6
**Confidence:** 3
**Correctness:** Yes.
**Clarity:** Yes.

**Review:**

## Pros
1. The release of REFUSE and the REFUSE-BENCH datasets under an open-source license promotes transparency and encourages further research and collaboration in the field.
2. The method description is clear.

## Cons
1. Figure 1 and the text overlap.
2. Missing details: The authors does not explain the meaning of `Error` in Table 4.
3. The results in the paper (Table 4) is somewhat rough: First, it only compares with the two existing baseline methods. Second, it does not show the search results for cross-optimization options.
4. The authors does not analyze the reason jTrans also outperforms REFuSe on BinaryCorp. Over-engineer does not seems to be an enough explanation.

**Strengths:**

1. The release of REFUSE and the REFUSE-BENCH datasets under an open-source license promotes transparency and encourages further research and collaboration in the field.
2. The method description is clear.

**Additional Feedback:**

NA

**Documentation:**

Yes.

**Ethics:**

No.

**Limitations:**

Yes.

**Opportunities For Improvement:**

1. Provide more detailed explanations about the missing results.
2. Modify the layout of the paper.

**Relation To Prior Work:**

Yes.

**Summary And Contributions:**

This paper is focused on the topic of binary function similarity detection (BFSD) in the context of machine learning for binary analysis. The authors present a benchmark dataset for BFSD and introduce a new baseline model called REFuSe. They compare REFuSe with two representative models, GNN and jTrans, and discuss the limitations and challenges in feature engineering and model scalability. The paper also addresses the importance of labeling details and the impact of data choice on evaluation and comparison.

---

> ### Author Rebuttal · Authors · 2024-08-15
>
> **C1**: We will fix this typographical error, thank you.
>
> **C2**: Thank you for this typo catch from our submission time. The table entry should have been "0.08", about 8.4$\times$ lower MRR than REFuSe.
>
> **C3**: Please see the all-author rebuttal on three new rows for Table 4 to help alleviate this concern.
>
> To clarify, Table 4 is a cross-optimization search in that it includes all optimization levels of the code in one large search pool. This makes the task harder as the number of alternative functions to be compared against. In all cases, we are searched for a function across different optimization levels and compilers.
>
> Doing only pairwise optimization tests is also problematic in a real-world usage context. First, we often never know what optimization levels are used, e.g., the MOTIF dataset is real-world malware with no such information. Second, compiling projects alone creates inequity in unique function counts (see Assemblage paper, Table 4). Applying filtering and using real optimization like function-inlining further complicates this, making pairwise cross-optimization levels difficult to interpret. For this reason, we do not recommend them, and we will clarify that in revision.
>
> **C4**: BinaryCorp is the dataset used/introduced by the jTrans paper, and thus represents the exact same population that jTrans was trained giving it an advantage (nothing is "out of distribution"). Given that reverse engineering a single executable is an effort measured in hours to weeks [A,B], it is not possible to perform a detailed analysis about why specific functions perform better/worse for a given method.
>
> In addition, we note that part of the intrinsic value of our benchmark is that we can identify that jTrans performs well on its BinaryCorp dataset, but does not generalize to new datasets. Since generalization is a key necessity in real-world usage (malware authors are an adaptive adversary that will alter their code to evade detection), this should be the expectation.
>
> A.  A. Mohaisen and O. Alrawi, “Unveiling Zeus: Automated Classification of Malware Samples,” in Proceedings of the 22Nd International Conference on World Wide Web
>
> B. Daniel Votipka, Seth Rabin, Kristopher Micinski, Jeffrey S. Foster, and Michelle L. Mazurek. 2019. An Observational Investigation of Reverse Engineers' Process and Mental Models. In Extended Abstracts of the 2019 CHI Conference on Human Factors in Computing Systems
>
> We thank the reviewer for their valuable questions and hope that the above has satisfyingly answered them. Please let us know if there is anything else we can clarify.

---

### Official Review · Reviewer_wxy9 · 2024-08-01
**binary function similarity detection based on binary vector**

**Rating:** 7
**Confidence:** 4
**Clarity:** The paper is well-written and the sup…

**Review:**

This work addresses a critical shortcoming of recent research, due to the cherry-picking of binaries compiled with a limited set of compiler options/flags. The authors identify 5 common discrepancies (identified as C1-C5) in the dataset constructions and mitigated them by carefully splitting the large dataset.

**Strengths:**

*Use of large dataset for Windows generated using Assemblage
*Consideration of the context of the executable, including compiler flags, macro evaluation

**Additional Feedback:**

I did not fully understand the ranking you talk in the work. Can you provide more description on Ranking? I believe the MRR was chosen to showcase the accuracy of the labeling. If that is the case why other standard methods (confusion metrix, precision, recall etc) are not considered for the evaluation?

Can you provide a complementary evaluation on some of the other DLs such as CNN/RNN and/or other transformers?

**Correctness:**

The authors have created a larger dataset including the context which should provide more detailed features in return. The proposed evaluation seems to be correct, according to my knowledge.

**Documentation:**

Is the Dataset publicly available? Is REFuSe publicly available?

**Ethics:**

No violation observed

**Limitations:**

Considering the stream of binary as opposed to traditional relational graphs such as AST/CFG might have its own cons. Probably this is why it is hard to distinguish between the functions like vec<bool> against vec<int>. As correctly identified in the paper, this requires future attention.

**Opportunities For Improvement:**

I like the paper overall. The paper addresses many existing shortcomings.

**Relation To Prior Work:**

The authors have clearly mentioned the prior work against the identified weakness, and on the popular datasets.

**Summary And Contributions:**

This work introduces a new benchmark dataset, REFUSE-BENCH, for much-needed Windows executables. The dataset fulfills the requirements such as a dataset that reflects the real-world binaries including inter-procedural relations, a simple baseline that operates on binary stream without pre-processing them. The proposed baseline REFuSe is evaluated against GNN and jTrans, and  it is perform fairly well on 4 out of 5 datasets.

---

> ### Author Rebuttal · Authors · 2024-08-15
>
> >Considering the stream of binary as opposed to traditional relational graphs such as AST/CFG might have its own cons...
>
> We agree entirely and believe that future work using more domain knowledge like AST/CFG will outperform REFuSe. Part of the value of our work is identifying that these useful tools are not being effectively leveraged to their full potential today, and the size of our dataset will spur research into addressing the scalability challenges with such domain knowledge approaches as well.
>
> >Is the Dataset publicly available? Is REFuSe publicly available?
>
> We have received our sponsor's approval to make the dataset and code publicly available, and will host it on GitHub. The supplemental material had all the REFuSe code and the Assemblage "recipe" to create the dataset (it is too large to make as an attachment alone).
>
> >Can you provide more description on Ranking?
>
> The supplemental material in Appendix A.2 includes a more detailed description of MRR.
>
> We will add this short note to the main paper: The MRR can be interpreted as the average retrieval of how many $k$ items need to be retrieved to find the first relevant item. For example, an MRR of 0.6 means that, on average, 1/0.6 = 1.67 items need to be searched to find a relevant match, whereas an MRR of 0.01 would mean searching 1/0.01 = 100 items on average. Since RE tasks are only impacted by the first relevant result, MRR is the most directly correlative measure of real-world use, whereas Recall and Precision @k are impacted by the number of relevant items. This is problematic when different items have different denominators, which happens due to the filtering and compiler optimizations that can occur, as discussed in Table 3.
>
> Separately, we note that jTrans and some other papers circumvent this by using non-standard definitions of Recall@k. We have used this same definition in the all-author rebuttal as it is computationally more feasible to get for all methods under consideration.
>
> >Can you provide a complementary evaluation on some of the other DLs such as CNN/RNN and/or other transformers?
>
> REFuSe is a type of CNN, though it is unusual in its wide receptive window. We are not sure if we can run an RNN/other DL in the remaining time of the rebuttal, but we will make an honest effort to do so as we run the other requested experiments.
>
> We hope this satisfies your questions and we appreciate the review. Please let us know if there is anything else we can help clarify.

---

> > ### Author Rebuttal · Authors · 2024-09-01
> >
> > We are at the end of the rebuttal, and unfortunately, we are answering some of the other questions reviewers had dominated our computing resources.
> >
> > We have started an experiment using a Transformer as a replacement for the convolutional layers in REFuSe. The transformer appears to meaningfully degrade performance; on the Common Libraries dataset, performance drops to an MRR of (0.526, 0.536), and a significant drop occurs attempting to generalize to Linux binaries with an MRR of (0.078, 0.106). This is an interesting result that we will be sure to include. Thank you for the suggestion.

---

### Author Rebuttal · Authors · 2024-08-15

Multiple reviewers have noted that they would like more baseline methods in the comparison results. We will add the following exposition and additional results to the camera-ready manuscript.

First, as the jTrans paper itself noted[41], the computational cost of most methods is prohibitively exorbitant (e.g., graph matching is NP-hard). Second, most alternatives require proprietary software or do not share the code (and rely on highly technical domain-specific feature engineering). For example, the PalmTree algorithm relies on software called BinaryNinja. Third, the Marcelli work [33] did a large analysis of binary function similarity methods and identified the GNN as the overall most effective. Given the enormous cost of performing this analysis, we believe this is a well-justified selection.

Still, we want to maximally satisfy the concerns raised despite the impossibility of using most prior works. We have conducted two sets of additional experiments to satisfy this concern.

First, we have re-trained GNN on the Assemblage dataset from scratch (GNN-A), eliminating differences in training data as a factor. The jTrans algorithm does not release its training code, which uses a process different from that of its fine-tuning code. Thus we have fine-tuned jTrans (jTrans-F) on the Assemblage data. In both cases performance still lags behind significantly, and helps isolate that these methods are in a sense over-engineered as they perform worse even when using the same data.

|  | Assemblage | MOTIF | Common Libraries | Marcelli Dataset | BinaryCorp |
|:---:|:---:|:---:|:---:|:---:|:---:|
| GNN-A | (0.267, 0.288) | (0.613, 0.622) | (0.109, 0.135) | 0.14 | (0.089, 0.117) |
| jTrans-F | 0.475 | 0.069 | 0.677 | 0.139 | 0.551 |
| BSim | (0.158, 0.187) | (0.143, 0.150) | (0.434, 0.475) | (0.279, 0.334) | (0.288, 0.322) |

To strengthen this further, we have added a third experiment called BSim. BSim is a tool for function similarity search developed by the Ghidra reverse engineering tool (see  [here](https://htmlpreview.github.io/?https://github.com/NationalSecurityAgency/ghidra/blob/Ghidra_11.0_build/Ghidra/Configurations/Public_Release/src/global/docs/WhatsNew.html) ). It is specifically designed for finding similar functions and is intended to be invariant to changes in the architectural platform.

On Assemblage, BSim gets a max MRR of 0.187, 0.150 on MOTIF, 0.414 on Common Libraries, 0.334 on Marcelli, and 0.322 on BinaryCorp. These are well behind REFuSe (min 0.611, 0.678, 0.684, 0.676, 0.468 respectively). BSim is extremely expensive to extract -- 24 minutes of CPU compute time per file.

In short, BSim works by first disassembling bytes, then decompiling the assembly to an intermediate representation called ``p-code'', and then creating feature vectors from the p-code. This provides an additional example of how our simpler method is performing better than a domain knowledge-heavy system that was hand-designed for this task (there is no training BSim; it is all human expert extraction/processing).

We believe that such domain knowledge will eventually outperform REFuSe in the long term, but we argue our results show that such expertise is not being efficaciously used today.

In addition, Recall@k was requested as a metric and is included below in the same calculation as jTrans. Empty values are still running, but it is clear that the same conclusions are reached. REFuSe outperforms others on all datasets except BinaryCorp, which jTrans wins, but it also has the advantage of being trained specifically for BinaryCorp. This also includes the three new baselines.

| Model | Dataset | R@1 | R@2 | R@5 | R@10 |
|:---:|:---:|:---:|:---:|:---:|:---:|
| GNN | Assemblage | 0.238 | 0.265 | 0.295 | 0.318 |
| GNN | MOTIF | 0.499 | 0.592 | 0.709 | 0.782 |
| GNN | Common Libraries | 0.085 | 0.104 | 0.135 | 0.159 |
| GNN | Marcelli Dataset-1 | 0.44 | 0.53 | 0.625 | 0.70 |
| GNN | BinaryCorp | 0.124 | 0.141 | 0.165 | 0.183 |
| GNN-A | Assemblage | 0.236  | 0.264 | 0.296 | 0.320 |
| GNN-A | MOTIF | 0.568 | 0.619 | 0.671 | 0.695 |
| GNN-A | Common Libraries | 0.085 | 0.106 | 0.137 | 0.161 |
| GNN-A | Marcelli Dataset-1 | 0.065 | 0.115 | 0.2 | 0.33 |
| GNN-A | BinaryCorp | 0.079 | 0.091 | 0.109 | 0.123 |
| jTrans | Assemblage | 0.221 | 0.288 | 0.338 | 0.376 |
| jTrans | MOTIF | 0.043 | 0.063 | 0.096 | 0.128 |
| jTrans | Common Libraries | 0.510 | 0.600 | 0.689 | 0.748 |
| jTrans | Marcelli Dataset-1 | 0.068 | 0.076 | 0.105 | 0.132 |
| jTrans | **BinaryCorp** | **0.625** | **0.700** | **0.775** | **0.826** |
| BSim | Assemblage | 0.143 | 0.164 | 0.175 | 0.181 |
| BSim | MOTIF | 0.087 | 0.103 | 0.135 | 0.179 |
| BSim | Common Libraries | 0.389 | 0.442 | 0.487 | 0.515 |
| BSim | Marcelli Dataset-1 | 0.231 | 0.280 | 0.336 | 0.371 |
| BSim | BinaryCorp | 0.251 | 0.291 | 0.329 | 0.354 |
| REFuSe | **Assemblage** | **0.567** | **0.610** | **0.664** | **0.700** |
| REFuSe | **MOTIF** | **0.623** | **0.675** | **0.744** | **0.792** |
| REFuSe | **Common Libraries** | **0.648** | **0.683** | **0.727** | **0.757** |
| REFuSe | **Marcelli Dataset-1** | **0.629** | **0.679** | **0.727** | **0.752** |
| REFuSe | BinaryCorp | 0.380 | 0.426 | 0.655 | 0.674 |

---

> ### Author Response · Authors · 2024-08-15
> **Full Recall@K results in**
>
> We are posting to let you know that the full Recall@K results are in and, as expected, fully align with the MRR results.  The original table has been updated with the complete numbers.

---

> > ### Author Rebuttal · Authors · 2024-08-19
> >
> > Dear reviewers, the JTrans-F recall was missing from the above table. Please see it below, though no changes are notable from the MRR results.
> >
> > | Dataset    |  R@1  |  R@2  |  R@5  | R@10 |
> > | ------------|--------|--------|--------|------- |
> > | Assemblage | 0.347 | 0.427 | 0.531 | 0.613 |
> > | MOTIF      | 0.030 | 0.040 | 0.050 | 0.083 |
> > | Common Lib | 0.577 | 0.682 | 0.797 | 0.852 |
> > | Marcelli   | 0.065 | 0.089 | 0.130 | 0.166 |
> > | BinaryCorp | 0.557 | 0.618 | 0.684 | 0.732 |

---

### Decision · Program_Chairs · 2024-09-26

**Decision:**

Accept (Poster)

**Comment:**

This paper builds a new benchmark, REFuSe-Bench, for binary function similarity detection. Most reviewers provide positive score for this paper. The only concerns from the reviewer after rebuttal is the the limited pool size. The authors have conducted experiments in the last minutes. AC read the rebuttal and felt that the concerns have been addressed. Thus, AC recommend the acceptance for this paper.